# Rehydration Activity of High-Temperature Calcined Recycled Sand Autoclaved Aerated Concrete

**Xiuli Yang** **, Renmiao Zhu and Bin Xu** *

School of Civil and Architecture Engineering, Nanchang Institute of Technology, Nanchang 330099, China
* Correspondence: xmq418@163.com

**Abstract:** Autoclaved aerated concrete is an excellent thermal insulation wall material, but with a large amount of waste. This paper describes the high-temperature activation and rehydration activity of waste cement–lime–sand autoclaved aerated concrete (SAAC) and discusses the high-temperature phase transition of SAAC. SAAC calcined at 750 °C was confirmed to be a metastable and amorphous state, which could hydrolyze $Ca^{2+}$ ions with reactivity in water. The conductivity curve demonstrates that the concentration of ions in the suspension decreases rapidly at 150–250 min, and the hydration reaction turns dominant at this time. The hydration heat curve also displays a hydration exothermic peak at 2.5 h. In addition, the conductivity measurement of suspension and the exothermic measurement of hydration reaction proves that SAAC calcined at 750 °C has a hydration activity and can rehydrate with $SiO_2$ in the system. The rehydration activity was verified by replacing 30% cement in the standard test block with calcined SAAC because the calcined SAAC at 750 °C has high hydration activity, and its activity index reached 89.58%. Fly ash is a commonly used cement admixture at present. Hence, the SAAC calcined at 750 °C and the fly ash were used to replace 30% of the cement in the cement test block, respectively. The results of this comparative experiment vividly showed that the reaction activity of SAAC calcined at 750 °C was higher than that of fly ash. Therefore, according to this research, SAAC has activity after calcination at 750 °C and can be hydrated again.

**Keywords:** autoclaved aerated concrete; electrical conductivity; hydration heat; rehydration activity; strength activity index; calcined

## 1. Introduction

Autoclaved aerated concrete (AAC) is a lightweight concrete with a porous structure. Its dry density is generally 400–800 kg/m³, 1/3–1/5 of ordinary concrete, and its compressive strength is usually 2–7 MPa [1]. The porous structure of AAC helps it produce excellent heat and sound insulation effects [2–4], and these characteristics make it an ideal wall material [5,6]. In addition, AAC has low density and a light weight, which can reduce both the self-weight of buildings and the construction cost. As a new type of filling wall material, it has been widely used in the market [7–9]. However, due to its low strength [10], about 5–10% of AAC blocks will inevitably be damaged in production. Furthermore, with the process of urbanization and reconstruction of old buildings, a large amount of waste aerated concrete will also be produced. The waste AAC is harmful to the environment in a direct or indirect way, such as occupying the land and polluting the water, air and soil. Therefore, it is necessary to find a new method to recycle the waste AAC.

Many scholars have studied recycled waste aerated concrete. Shui [11] grinded the waste aerated concrete to particles finer than 75 μm, and then heated them to 900 °C. Then fly ash was added to prepare new cementitious materials to study the interaction between fly ash and DAAC. Ullrich [12] added $CaCO_3$ into the waste aerated concrete, and adapted the molar $CaO/SiO_2$ (C/S ratio) in the range of 2 to 2.5 to synthesize Belite binders. Several factors influencing the reaction kinetics and the evolution of the phase composition were

investigated in his research. Chindaprasirt [13] used recycled lightweight aggregate from waste autoclaved aerated concrete blocks to make lightweight pervious concrete (LWPC). The LWPCs had a low density of 775–900 kg/m$^3$ and low thermal conductivity coefficient of 0.15–0.27 W/m·K. It is suitable for use as thermal insulating concrete.

In order to deal with the waste AAC, some Chinese enterprises usually use it as lightweight aggregate [14] or apply it in thermal insulation roofing after crushing. The roof insulation material has many shortcomings, such as low strength, easy damage, inconvenient maintenance, and short service life, and is gradually replaced by new materials. The waste AAC can still be discarded and stacked optionally.

In this paper, the calcination temperature was selected according to the DSC-TG curve, and the phase of the SAAC was analyzed by XRD after calcination at different temperatures. It is revealed that the hydration products of SAAC gradually lose the interlayer water and then the hydroxyl group during the heating process. At about 750 °C, the hydration product tobermorite changes from a crystalline state to an unstable amorphous state. If the calcination temperature continues to rise, the amorphous transition phase will be reconstructed and converted to a crystalline state again, and wollastonite will be generated. In order to verify that the hydration product of SAAC calcined at 750 °C is metastable, the sucrose method is used for chemical titration. The results confirmed that after calcination at 750 °C, a large amount of CaO was hydrolyzed from the aqueous solution. In the conductivity test of suspension, the conductivity of SAAC calcined at 750 °C is higher than that at other temperatures within 150 min. It is also proved that there are many ions in the solution at this time, which is consistent with the results of the determination of CaO by the sucrose method.

Through various experimental methods and tests, it is proved that SAAC has ions dissolved after calcination at 750 °C, and the Ca$^{2+}$ is dissolved. It is also confirmed that calcination will cause the molecular bonds in the crystal to break, the crystal structure to collapse, and the transition phase to be unstable. Hydrolyzed CaO will react with unreacted SiO$_2$ in SAAC, which is a common calcium silicate hydration reaction. Measurements using a microcalorimeter show that there is hydration heat release after 2.5 h, which means a hydration reaction occurs. Then the strength index method is used to prove the activation performance after calcination at 750 °C. These results show that the waste SAAC calcined at 750 °C–825 °C have strong rehydration activity, and the strength activity index reached 87.7% after 28 days. Thus, this study reveals that waste SAAC calcined at a certain temperature has rehydration activity, and it can be recycled.

## 2. Materials and Experimental Methods

### 2.1. Material

There are mainly two systems used to produce AAC based on different raw materials. One is cement–lime–sand autoclaved aerated concrete (SAAC), and the other is cement–lime–fly ash autoclaved aerated concrete (FAAC). This experiment was carried out with the SAAC produced by an AAC manufacturer in Nanjing, China. The SAAC was milled into powder and dried at 105 °C in an oven. The median diameter D50 of the SAAC powder was 45.49 μm and for D98 it was 199.58 μm. The chemical composition of the SAAC is given in Table 1. The loss on ignition (LOI) was 8.52%, which contained mainly hydroxyl and interlayer water.

**Table 1.** The chemical composition of SAAC.

| Precntage by Mass (%) | | | | | |
|---|---|---|---|---|---|
| SiO$_2$ | Al$_2$O$_3$ | Fe$_2$O$_3$ | CaO | Other | LOI |
| 57.34 | 5.16 | 1.9 | 26.61 | 0.47 | 8.52 |

Prepared according to the national standard, the cement mortar test block was used to study the activity index of the admixture. The cement was ordinary Portland cement from

Jiangnan Xiaoyetian Cement Co., Ltd. in Nanjing, China, and the standard sand was from ISO Standard Sand Co., Ltd. in Xiamen, China.

### 2.2. The Measurement Method of ef-CaO

The solubility of CaO in water is so small that direct titration with HCL will lead to a large deviation. However, adding sucrose will help CaO form "calcium sucrose" with higher solubility, and then titrate the content of CaO in calcium sucrose with HCl. Finally, the content of ef-CaO was calculated by the amount of HCl standard solution consumed in titration. This is the principle of measuring ef-CaO by the sucrose method. Equation (1) describes the formation of calcium sucrose from CaO and sucrose, while Equation (2) is the process of titration of calcium sucrose with hydrochloric acid.

$$C_{12}H_{22}O_{11} + CaO + 2H_2O \rightarrow C_{12}H_{22}O_{11} \cdot CaO \cdot 2H_2O \tag{1}$$

$$C_{12}H_{22}O_{11} \cdot CaO \cdot 2H_2O + 2HCl \rightarrow C_{12}H_{22}O_{11} + CaCl_2 + 3H_2O \tag{2}$$

Before the experiment, we prepared a dry sample, accurately weighed 0.5 g of the sample, and put it in a 250 mL conical flask. Then, we weighed 4 g of sucrose and covered it on the surface of the sample. We added 15–20 glass beads, added 50 mL of boiled pure water, plugged and then shook it for 20 min. Later, we opened the bottle stopper, washed the bottle stopper and bottled the wall with boiled pure water, and added the phenolphthalein indicator. A titrate with 0.5 N hydrochloric acid solution until the pink color of the solution disappears and does not reappear within 30 s, marks the end of the experiment. The volume of hydrochloric acid consumed in this process is $V$. The content of effective calcium oxide can be calculated by Equation (3), and then the calcium ion dissolution can be calculated. In Equation (3), $N$ is the equivalent concentration of HCl, and $M$ is the mass of the sample.

$$CaO\% = \frac{V \times N \times 0.02804 \times 100}{M} \tag{3}$$

### 2.3. The Method of Strength Index

The intensity activity index method is based on the national standards GB/T 12957-2005 [15] and GB/T 17671-2021 [16]. The test methods provided by Chinese National Standard GB/T 12957-2005 and American Standard ASTM C311-18 are similar because both of them are based on the strength activity index. The size of the test block in the experiment is 40 mm × 40 mm × 160 mm. We prepared 6 samples, and measured their compressive strength after curing for 28 days. The average value of 6 samples was regarded as the compressive strength of this group. In terms of the strength activity index method, a certain amount of mixed material is usually added as a substitution for cement. The compressive strength at 28 d of blocks can determine the activity index.

The cementing material in the standard test block is ordinary Portland cement. In the reference block, the cementing material is 70% cement and 30% other materials. The 30% material added in the reference block is the material that needs to be tested for the activity index. The activity index can be determined as follows:

$$A = \frac{R_1}{R_0} \tag{4}$$

where $A$ is the strength activity index, $R_1$ is the compressive strength of blocks containing mixed materials at one age (MPa), and $R_0$ is the compressive strength of blocks without mixed materials at the same age (MPa).

## 3. Experimental Results and Discussion

### 3.1. Material Analysis

Figure 1a shows the X-ray diffraction pattern of the SAAC. The pattern illustrates that the SAAC contains hydration products such as C-S-H gel and tobermorite ($Ca_5Si_6O_{16}(OH)_2 \cdot 4H_2O$), and part of the $SiO_2$ did not take part in the reaction.

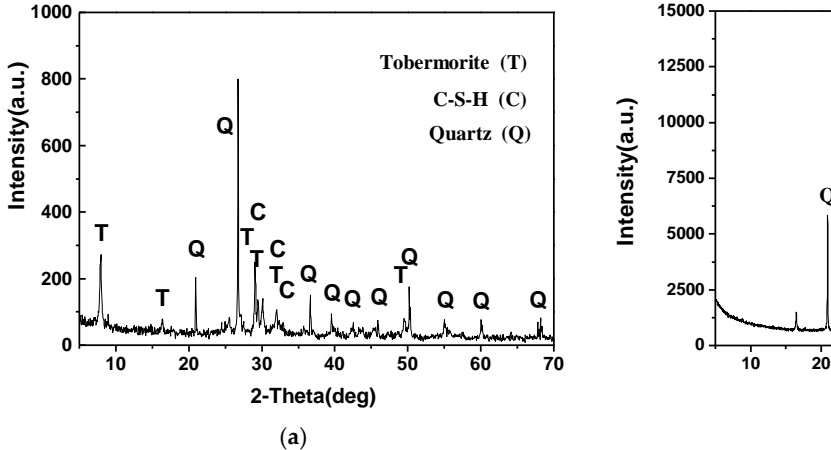 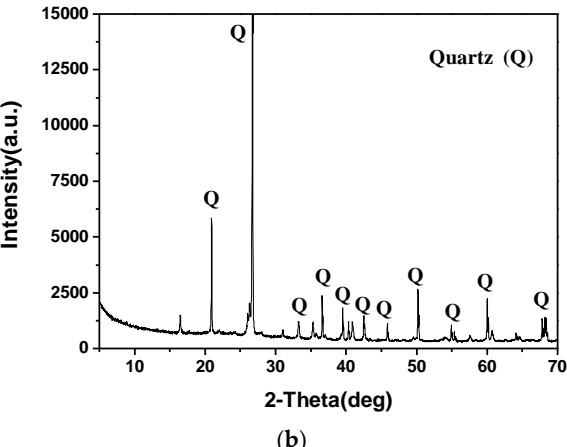

(**a**) 　　　　　　　　　　　　　　　　　　　　　　(**b**)

**Figure 1.** X-ray diffraction pattern of SAAC and acid-insoluble substance. (**a**) X-ray diffraction pattern of SAAC; (**b**) X-ray diffraction pattern of acid-insoluble substance.

Take 0.5 g of SAAC powder, dissolve it with $70 \pm 2$ °C diluted HCl (1 + 10) for 20 min, and filter and wash until the chloride reaction disappears. The hydration products react with hydrochloric acid, while $SiO_2$, $Al_2O_3$, $Fe_2O_3$ and some impurities do not react with hydrochloric acid. Burn the filter paper and residue at 900–1000 °C to a constant weight, which is called an "acid-insoluble substance". The acid-insoluble substance of SAAC is 27.7%, mainly the unreacted $SiO_2$. Figure 1b shows the XRD diffraction pattern of SAAC acid-insoluble substances. It can be seen that most of the acid-insoluble substances are unreacted $SiO_2$, accounting for 83.5% of the residue, which can react with the calcareous phase after SAAC activation.

Figure 2 presents the TG-DTA curve of the SAAC. As shown in Figure 2, endothermic peaks of SAAC at 76 °C, 146 °C, 571 °C, 745 °C and 820 °C, respectively and a strong exothermic peak at 856 °C can be observed. The molecular water of tobermorite is lost at about 100 °C. It can be seen from the TG curve that the weight loss of 100–300 °C is 9.61%, indicating that the interlayer water is gradually lost at this time. There are three endothermic peaks at 571 °C, 745 °C and 820 °C, indicating that some substances decompose at this temperature. It can be inferred that CSH gel and tobermorite were decomposed [17]. The exothermic peak appears without weight loss at 856 °C, which can be attributed to the crystal phase transition of the sample.

In order to analyze the phase change of SAAC at high temperatures, the calcination temperature was determined according to the results of differential thermal analysis (DTA curve). When decomposition occurs, it needs to absorb energy and the chemical bond breaks. When combining, chemical bonds are formed and energy is released. The phase change usually occurs when it is exothermic or endothermic, and the phase stabilized after the peak. The calcination temperatures were determined to be 150 °C, 500 °C, 600 °C, 750 °C, 825 °C, 900 °C and 1000 °C, respectively, and the SAAC was calcined in a resistance furnace for 1 h.

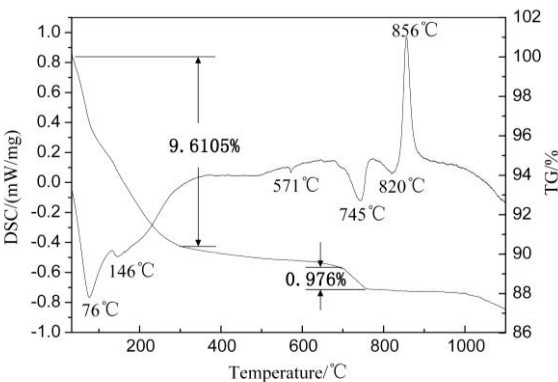

**Figure 2.** TG−DTA curve of the SAAC.

Figure 3 shows the diameter distribution of SAAC calcined at different temperatures. Table 2 is the specific measured data. According to the result, when heated to 300 °C, the particle size decreases, and the D50 of SAAC decreases from 45.49 μm to 36.69 μm, and D98 reduced from 199.58 μm to 167.71 μm. When the calcination temperature reaches 750 °C, D50 changes very little, and D98 is further reduced to 133.05 μm. It is shown that the SAAC of large particles is further reduced at this time, and when the calcination temperature rises to 1000 °C, the large particles of SAAC will shrink a little.

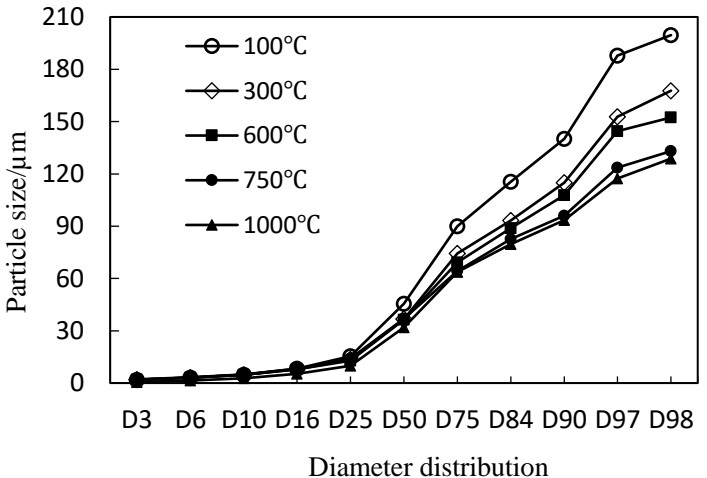

**Figure 3.** The diameter distribution of SAAC calcined at different temperatures.

**Table 2.** The diameter distribution of SAAC calcined at different temperatures.

| Diameter Distribution | 100 °C | 300 °C | 600 °C | 750 °C | 1000 °C |
|---|---|---|---|---|---|
| D50 | 45.49 | 36.69 | 36.39 | 36.98 | 32.04 |
| D98 | 199.58 | 167.71 | 152.35 | 133.05 | 128.76 |

### 3.2. The XRD Pattern of SAAC at Different Calcination Temperatures

Figure 4 shows the X-ray diffraction patterns of SAAC obtained at different calcination temperatures. As displayed in Figure 4a, the characteristic peaks of tobermorite and quartz are narrow with high intensity, and all diffraction peaks are clear and complete. The characteristic peak of 11.3 Å, 5.48 Å, 3.08 Å, 2.98 Å, 2.82 Å, 2.08 Å, 1.842 Å of tobermorite is clearly visible, indicating that tobermorite crystal has high crystallinity.

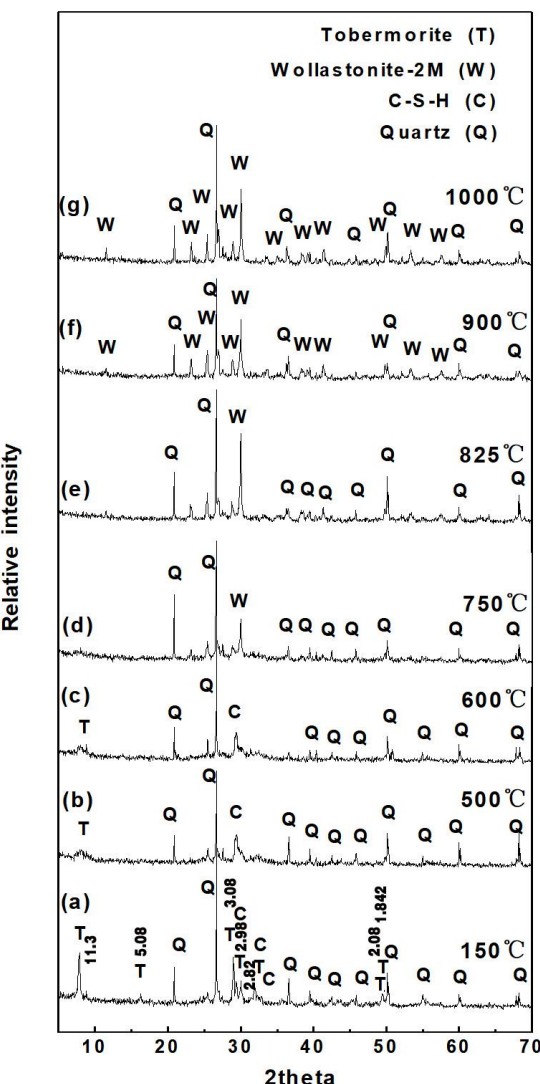

**Figure 4.** The X-ray diffraction pattern of SAAC at different calcination temperatures. (**a**) 150 °C; (**b**) 500 °C; (**c**) 600 °C; (**d**) 750 °C; (**e**) 825 °C; (**f**) 900 °C; (**g**) 1000 °C.

As shown in Figure 4b,c, most of the diffraction peaks of the tobermorite disappear after being calcined at 500 °C and 600 °C, and only the diffraction peaks of 11.3 Å (002) are left with lower intensity. In Figure 4d, all of the diffraction peaks of tobermorite disappeared and the remaining peaks were the characteristic peaks of quartz. It suggests that the crystal structure of tobermorite is broken, the hydroxyls are lost and the structure tends to be in a disordered, metastable, amorphous state at 750 °C. Figure 4e,f demonstrates that SAAC remains an amorphous state at 825 °C and the new characteristic diffraction peaks of wollastonite appear as the calcination temperature reaches 900 °C. The DTA curve (Figure 2) has an exothermic peak at 856 °C, which illustrates 856 °C is a phase-transition temperature. The hydration product tobermorite in SAAC changes or recombines to be wollastonite at the transition temperature. As the calcination temperature rises to 1000 °C, the diffraction peaks become sharp and complete, which suggests that the 2 M-wollastonite phase is obtained as compared with the standard JCPDS card (27-0088).

*3.3. Hydrolysis Performance of Heated SAAC at 750 °C*

The main hydration product in SAAC is tobermorite, a hydrated calcium silicate mineral. The molecular formula is $Ca_5Si_6O_{16}(OH)_2·4H_2O$. The structure of tobermorite with am11.3 Å diffraction peak is orthorhombic. The arranged silicon oxide tetrahedral chains repeatedly form the layers, and the silicon chains are connected by calcium ions

between the layers. Four molecules of water are distributed between the layers, and two hydroxyl groups are connected with the silica tetrahedron.

The binding force of interlayer water is weak. When the temperature rises, the interlayer water gains energy, its movement intensifies, and it is easy to take off. In the TG-DTA curve, there is a large mass loss at 100–300 °C, which is the interlayer water loss. The hydroxyl group is connected by molecular bonds, which requires a large amount of energy to break. When the heating temperature reaches 750 °C, the hydroxyl bond breaks and the hydroxyl is removed. There is also a 0.976% mass loss in the TG-DTA curve, which is caused by the removal of the hydroxyl.

The molecular water and the interlayer water of tobermorite are gradually lost at about 50–300 °C. The hydroxyls are lost when calcined at about 745–750 °C. The period of calcination can be described with regard to the reaction equation as follows:

$$Ca_5Si_6O_{16}(OH)_2 \cdot 4H_2O \overset{50-300°C}{\rightarrow} Ca_5Si_6O_{16}(OH)_2 + 4H_2O \tag{5}$$

$$Ca_5Si_6O_{16}(OH)_2 \overset{745°C}{\rightarrow} Ca_5Si_6O_{17} + H_2O \tag{6}$$

Equation (5) describes a dehydration process of tobermorite. Four interlayer waters were lost when the tobermorite was heated to 300 °C. Equation (6) describes a dehydroxylation process. The "tobermorite" without molecular water and hydroxyl is not the original crystal structure. $Ca_5Si_6O_{17}$ is called dehydroxylated tobermorite, which is an amorphous metastable state. It can also be seen from the XRD diagram at 750 °C that the crystal plane diffraction of tobermorite disappears. It further proves that the crystal structure of tobermorite collapsed after the loss of interlayer water and hydroxyl. It can hydrolyze $Ca^{2+}$ ions in water and has the ability of a rehydration reaction.

The solubility of CaO in water is so small that the direct titration with HCL will cause a large deviation. Adding sucrose can help CaO form "calcium sucrose" with higher solubility, and then titrate the content of CaO in calcium sucrose with HCL. Finally, we calculated the content of ef-CaO by the amount of HCL standard solution consumed in titration. This is the principle of measuring ef-CaO by use of the sucrose method. The ef-CaO of SAAC calcined at different temperatures was measured by the sucrose method, described in Section 2.2. The obtained results of the ef-CaO content are given in Figure 5. The ef-CaO content at 750 °C and 825 °C is much higher than other calcination temperatures.

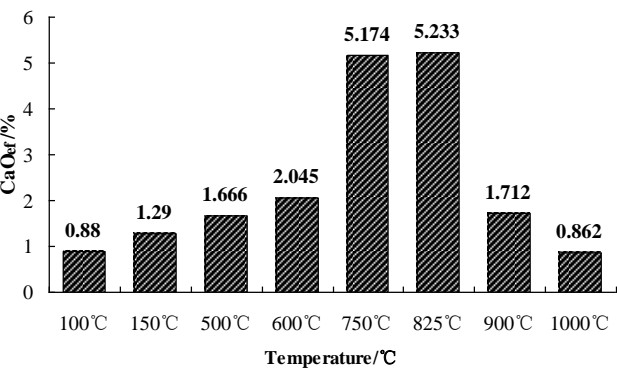

**Figure 5.** The ef-CaO content of SAAC at different calcination temperatures.

When the calcination temperature of the aerated concrete reaches 750 °C, part of the Si-O-H bonds will break and the hydroxyl will be removed, and the crystal structure of the main hydration products will be destroyed, which is in a disordered amorphous state. It can be seen from the results that a large amount of $Ca^{2+}$ ions dissolved from SAAC calcined at 750 °C. However, no characteristic peak of CaO was found in the XRD of SAAC calcined at 750 °C. This is because the crystal structure of tobermorite is destroyed, but calcium oxide is metastable in the structure. Calcium oxide is in an amorphous state without the X-ray diffraction phenomenon of the crystal surface. Metastable calcium oxide has

high activity and can be hydrolyzed from the structure to produce calcium hydroxide. After reaching the dissolution equilibrium, the aqueous solution shows strong alkalinity. Although the solubility of calcium oxide is very low, the hydrolyzed calcium oxide has strong chemical reactivity.

When the calcination temperature reaches 900 °C, the aerated concrete undergoes a solid phase reaction, and the damaged tobermorite structure recombines to form a stable wollastonite crystal structure. The arrangement of atoms tends to be ordered, and calcium oxide exists in the structure in the form of molecular bonds, losing its hydrolytic activity.

### 3.4. Conductivity Measurement of SAAC Suspension

The conductivity change of aqueous solution can reflect the change of ion concentration in the solution [18,19]. The higher the ion concentration is, the greater the solution conductivity is. The SAAC powder calcined at different temperatures was prepared into 1 g/L suspension, and the conductivity of the aqueous solution was measured continuously for 10 h with a conductivity meter. The conductivity meter was produced by INESA Scientific Instrument Co., Ltd. in Shanghai, China, and its model was Thunder magnetic DDS-307A. The conductivity was measured every minute from 0–30 min, every 5 min from 30–120 min, and every 10 min from 120–600 min.

Figure 6 shows the conductivity curve of SAAC suspension calcined at different temperatures. As shown in this figure, all curves demonstrate that the conductivity increases rapidly within 0–15 min. SAAC is a mixture, and many components will be slightly soluble in water, which will increase the ionic concentration and conductivity of water. The conductivity of SAAC calcined at 750 °C and 825 °C is significantly higher than that of others, indicating that more ions are dissolved in water. It also proves that the SAAC calcined at 750 °C and 825 °C has strong hydrolysis activity and releases a large number of $Ca^{2+}$ ions.

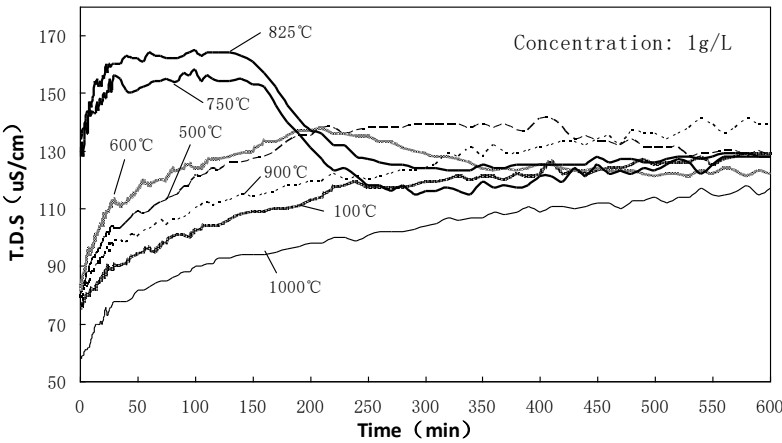

**Figure 6.** Conductivity of SAAC suspension calcined at different temperatures (0–10 h).

The conductivity curves of the samples calcined at 100 °C, 500 °C, 600 °C, 900 °C and 1000 °C are similar. As time went on, SAAC released ions constantly and the conductivity curve increased slowly until it finally approached a flat line. However, the curves of the samples calcined at 750 °C and 825 °C are different from other temperatures. The conductivity reached 150–160 us/cm after 15 min and before 150 min, respectively; then it reduced rapidly until it reached an equilibrium state at 250 min. When the ion dissolution rate was larger than the ion hydration reaction rate, the ion concentration or conductivity increased gradually; when the ion dissolution rate is equal to the ion hydration reaction rate, the conductivity changes a little; when the ion dissolution rate is smaller than the ion hydration reaction rate, the ion concentration or conductivity decreases gradually.

The ions of SAAC dissolved constantly, and meanwhile, the dissolved ions in the solution recombined to be the new hydration product within 15–150 min for the SAAC calcined at 750 °C and 825 °C. During this time, the ion dissolution rate was basically

in equilibrium with the ion recombination rate, and the ion concentration was constant. Over 150–250 min, the ion concentration of the solution decreased significantly; hence, the conductivity decreased as well. At this time, the ion hydration reaction in the solution dominated.

The change rule of conductivity reflected the hydration reaction capacity of SAAC. When SAAC is calcined at 100 °C, 500 °C, 600 °C, 900 °C and 1000 °C, the hydration process cannot be obviously observed. On the contrary, the SAAC calcined at 750–825 °C shows significant reaction activity. These samples had rehydration reaction capability and their hydration reaction would occur within 150–250 min.

### 3.5. Hydrated Heat of SAAC Calcined at 750 °C

The hydration process is an exothermic reaction. Hydrated heat could be determined by the micro-calorimeter. Heat is released during hydration, and the resultant rise in temperature is captured by the heat flow sensors within the micro-calorimeter [20]. Figure 7 is the heat of the hydration curve of the SAAC calcined at 750 °C. Figure 7a is the measured complete exothermic curve, and two exothermic peaks can be found. The first one appears at 0.1 h, which was caused by the heat release of SAAC dissolution (Figure 7b); the second one appears at 2.5 h, namely, 150 min, which was due to the heat release of the SAAC hydration reaction (Figure 7c).

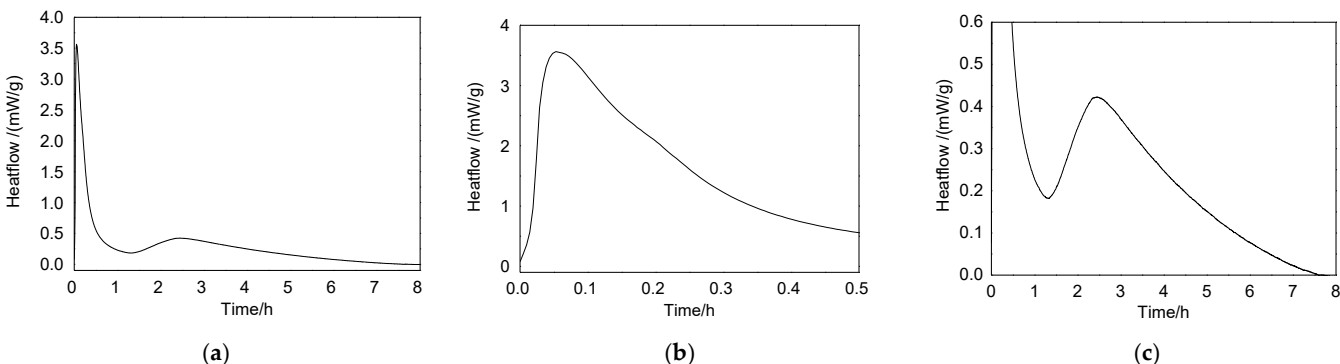

**Figure 7.** The heat of the hydration curve of SAAC calcined at 750 °C. (**a**) 0–8 h; (**b**) 0–0.5 h; (**c**) 0.5–8 h.

After the SAAC is mixed with water, due to the surface tension, the particle surface is not completely saturated. With the continuous wetting and dispersion of particles, the dissolution speed is accelerated and the heat release is rapid. The first peak of the heat of the solution appears at about 0.1 h, and the process is rapid, so the peak is high and narrow. After the hydration induction period, it enters the hydration period within 150 min (2.5 h), and the hydration reaction occurs and releases heat. However, the hydration reaction releases slight heat and lasts for a long time, so the peak is low and wide.

Heat of the hydration curve (Figure 7) also shows that the hydration reaction of the SAAC calcined at 750 °C will occur at about 150 min. The result is similar to the conductivity result. This proves that the conductivity curve and heat of the hydration curve are effective ways to monitor the occurrence time of the hydration reaction.

When the aerated concrete is calcined at 750 °C, the crystal structure of tobermorite is destroyed, calcium oxide can be hydrolyzed, and the hydrolyzed calcium oxide has potential chemical reactivity. There is a part of unreacted silica in SAAC, so the hydrolyzed calcium oxide can react with silica. This is also the main reason for the rehydration reaction of aerated concrete after calcination at 750 °C.

### 3.6. Strength Activity Index

SAAC calcined at 750 °C has been proved to have rehydration ability, and its activity can be measured by an intensity activity index. Prepare the sample and calculate the strength index according to the method described in Section 2.3. Ten different mixtures of

cement mortar blocks are described in Table 3. A0 is the standard cement mortar blocks, and Af is the cement mortar blocks by which 30% of the mass of the cement is replaced by fly ash. In the samples of A100, A150, A500, A600, A750, A825, A900 and A1000, SAAC calcined at different temperatures is used to replace the cement quality of 30%.

**Table 3.** The mixture components of cement mortar block.

| | Temperature | Fly Ash /g | SAAC /g | Cement /g | Sand /g | Water /g |
|---|---|---|---|---|---|---|
| A0 | | - | - | 450 | 1350 | 225 |
| Af | | 135 | - | 315 | 1350 | 225 |
| A100 | 100 °C | - | 135 | 315 | 1350 | 225 |
| A150 | 150 °C | - | 135 | 315 | 1350 | 225 |
| A500 | 500 °C | - | 135 | 315 | 1350 | 225 |
| A600 | 600 °C | - | 135 | 315 | 1350 | 225 |
| A750 | 750 °C | - | 135 | 315 | 1350 | 225 |
| A825 | 825 °C | - | 135 | 315 | 1350 | 225 |
| A900 | 900 °C | - | 135 | 315 | 1350 | 225 |
| A1000 | 1000 °C | - | 135 | 315 | 1350 | 225 |

Figure 8 shows the compressive strength of cement mortar blocks at 28 days. In Figure 8a, the strength activity index of blocks containing SAAC calcined at 100 °C and 150 °C is 60.2% and 65.5%, respectively. However, the strength activity index of samples containing SAAC calcined at 1000 °C is only 56.1%. From Figure 8a, we can see that the strength activity index of blocks containing SAAC calcined at 750 °C reaches 89.58%, indicating that SAAC calcined at 750 °C has strong rehydration activity. According to Chinese National Standard GB/T 12957-2005, if the mixed material substitutes for 30% cement of the standard cement mortar blocks, the compressive strength at 28 days of blocks containing the mixed material will be no less than 65% of standard cement mortar blocks. According to this research, SAAC calcined at 750 °C has excellent rehydration activity, and its compressive strength is higher than the national standard of Chinese cement mixtures.

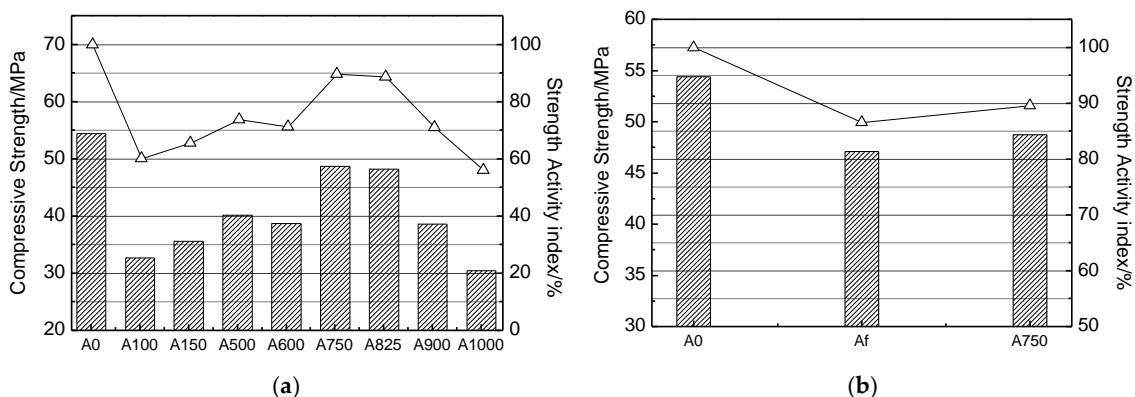

(a)  (b)

**Figure 8.** Compressive strength of cement mortar blocks at 28 days.

It is generally known that fly ash has higher pozzolanic activity and has been widely used in cement mixture. It can be seen from Figure 8b that the strength activity index of SAAC calcined at 750 °C is 3.04% higher than that of fly ash. This means that SAAC calcined at 750 °C also can be used as a mixture material, and its strength will be higher than fly ash.

The fineness of powder materials has a great influence on the strength. The powder with smaller particles has a larger specific surface area. The larger the contact area with the reactant, the more sufficient the reaction. However, from the experimental results of the strength index, the particle diameter is not the main factor affecting the strength. The 28-day compressive strength of A100, A150 and A500 increased from 32.75 MPa to 40.13 MPa, which is due to the removal of interlayer water in SAAC. With the same mass,

the effective reactants increase, slightly improving the strength. The compressive strength of A600 is 38.7 MPa, and the compressive strength of A750 increases to 48.73 MPa. At this time, the main reason for the strength increase is that CaO can be dissolved and has rehydration ability. However, the compressive strength of A900 and A1000 decreased to 38.63 MPa and 30.52 MPa again, which is due to the phase transformation and conversion to wollastonite phase after calcination to 900 °C. Although the particle size is the smallest at this time, wollastonite has no reactivity, which eventually leads to the strength reduction. It can be seen that when measuring the SAAC strength activity index, the particle size is not the main factor affecting the strength, but the tobermorite metastability and hydrolysis activity are the most influential factors.

## 4. Conclusions

(1)  The hydration products of SAAC mainly include C-S-H gel and tobermorite, and some unreacted $SiO_2$. The DTA curve of SAAC shows that there is a strong endothermic peak at 745 °C and a strong exothermic peak at 865 °C. By analyzing the phase changes of SAAC calcined at different temperatures, it was found that some Si-OH bonds were broken, hydroxyl groups were lost and crystal structure collapsed after calcination of tobermorite at about 750 °C. Tobermorite in SAAC tends to be disordered and amorphous.

(2)  After calcination at 750 °C, dehydroxylated tobermorite has hydrolytic activity, which can hydrolyze reactive $Ca^{2+}$ ions and react with sucrose. In the XRD of SAAC calcined at 750 °C, there is no diffraction peak of CaO, but $Ca^{2+}$ ions can be hydrolyzed. It further proves that after calcination at 750 °C, the crystal structure of the transition phase collapses and is metastable.

(3)  Monitoring the conductivity of suspension and measuring the hydration heat release are both effective methods to determine the occurrence of a hydration reaction. It was observed that SAAC calcined at 750 °C had a rehydration phenomenon, and the hydration reaction took place in about 2.5 h. Without adding other raw materials, the hydrolyzed $Ca^{2+}$ ions can react with unreacted $SiO_2$ in SAAC. It presents that calcining SAAC at 750 °C is actually an effective activation method.

(4)  After calcination at 750 °C, SAAC has reactivity and can be rehydrated. This paper replaced 30% of the cement in the standard mortar test block with the activated SAAC, and tested the activity index. The national standard specifies that the compressive strength at 28 days of blocks containing mixed material shall be no less than 65% the compressive strength of standard cement mortar blocks. From the activity index results, the activity index of the original SAAC was 60.2%, and the activity index of SAAC after calcination at 750 °C was 89.58%, a significant increase. Fly ash is a common cement mixture, but the activity index of SAAC after calcination at 750 °C is 3.04% higher than that of fly ash. From the perspective of applications, it is verified that SAAC calcined at 750 °C has the ability of rehydration, which also provides a new method for the reuse of waste SAAC.

**Author Contributions:** Experiment planning, X.Y.; experiment, X.Y. and R.Z.; data search, B.X.; experimental analysis, X.Y.; writing—original draft, X.Y. All authors have read and agreed to the published version of the manuscript.

**Funding:** This research was funded by Science and Technology Project of Jiangxi Provincial Department of Education (NO: GJJ201908), Key project of advantageous science and technology innovation team of Jiangxi province in 2017 ("5511"project) (NO: 20171BCB19001), The funding program for major disciplines academic and technical leaders of Jiangxi provincial in 2017 (NO: 20172BCB22022), The key projects of natural science foundation of Jiangxi Province (No: 20202ACBL204016), The key science and technology research project in Jiangxi province department of education (NO: GJJ151096), Innovation and Entrepreneurship Training Progran for College Students of Nanchang Institute of Technology (NO: 2021015).

**Data Availability Statement:** Not applicable.

**Conflicts of Interest:** The authors declare no conflict of interest.

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
