# Peer review of "Rehydration Activity of High-Temperature Calcined Recycled Sand Autoclaved Aerated Concrete"

_processes, doi:10.3390/pr11020422_

Round 1

Reviewer 1 Report

The paper lacks enough innovation, the analysis is not rigorous enough, and the logic is not good. The language also needs a lot of modification. The specific problems are as follows:

1) The abstract is not enough to introduce why and how to do the research, which can not reflect the novelty of this article.

2) The introduction mainly introduces autoclaved aerated concrete, but the current study and application status of recycled aerated concrete is seriously insufficient, and even some similar research authors have not introduced it(A. Ullrich, K. Garbev, B. Bergfeldt, In Situ X-ray Diffraction at High Temperatures: Formation of Ca2SiO4 and Ternesite in Recycled Autoclaved Aerated Concrete, Minerals 11(8) (2021); A. Ullrich, K. Garbev, U. Schweike, M. Kohler, B. Bergfeldt, P. Stemmermann, CaCl2 as a Mineralizing Agent in Low-Temperature Recycling of Autoclaved Aerated Concrete: Cl- Immobilization by Formation of Chlorellestadite, Minerals 12(9) (2022), et al.), indicating that the author lacks sufficient understanding of this field.

3) For the material part, the total amount does not reach 100%. At the same time, since it does not contain MgO, it is unnecessary to add it to the table.

4) In the results and discussion part, a large number of analyses lack rigor, and the experimental design is confusing. There are too many problems in this part, some of which are as follows:

P. 2 why do you want to study the dissolution rate, and what is the help for the follow-up research?

P.2  “The molecular waters and the interlayer waters in tobermorite lost at 50-300℃.” The expression is vague.

p.3  “As shown in Fig. 3(a), all diffraction peaks of SAAC calcined at 150ºC are clear and complete.” What is clear and complete?

p.4 88-89 It is crystal water, not hydroxyl, lost during heating.

p.4 The equation is difficult to correspond to the previous X-ray diffraction analysis.

p.6 183 Not Fig 1, but Fig. 6.

5) The conclusion is not refined enough.

Author Response

Response letter

We sincerely thank the editors and reviewers for their suggestions and comments, which we use to improve the quality of the manuscript. The reviewer's comments are displayed in normal font. Our reply is given in italics, and changes to the manuscript are given in blue text.

1) The abstract is not enough to introduce why and how to do the research, which can not reflect the novelty of this article.

Response: The abstract has been revised to give a general overview of the paper and emphasize that the research focus of this paper is the activation of SAAC.

2) The introduction mainly introduces autoclaved aerated concrete, but the current study and application status of recycled aerated concrete is seriously insufficient, and even some similar research authors have not introduced it(A. Ullrich, K. Garbev, B. Bergfeldt, In Situ X-ray Diffraction at High Temperatures: Formation of Ca2SiO4 and Ternesite in Recycled Autoclaved Aerated Concrete, Minerals 11(8) (2021); A. Ullrich, K. Garbev, U. Schweike, M. Kohler, B. Bergfeldt, P. Stemmermann, CaCl2 as a Mineralizing Agent in Low-Temperature Recycling of Autoclaved Aerated Concrete: Cl- Immobilization by Formation of Chlorellestadite, Minerals 12(9) (2022), et al.), indicating that the author lacks sufficient understanding of this field.

Response: The introduction has been modified and expanded. The added content is mainly the research status in this field.

3) For the material part, the total amount does not reach 100%. At the same time, since it does not contain MgO, it is unnecessary to add it to the table.

Response: The material table has been modified. Two columns have been added to the table, including LOI and other trace components. In addition, the particle size of SAAC powder is supplemented. The manufacturers of cement and sand used in the experiment are added.

4) In the results and discussion part, a large number of analyses lack rigor, and the experimental design is confusing. There are too many problems in this part, some of which are as follows:

  1. 2 why do you want to study the dissolution rate, and what is the help for the follow-up research?

P.2  “The molecular waters and the interlayer waters in tobermorite lost at 50-300℃.” The expression is vague.

p.3  “As shown in Fig. 3(a), all diffraction peaks of SAAC calcined at 150ºC are clear and complete.” What is clear and complete?

p.4 88-89 It is crystal water, not hydroxyl, lost during heating.

p.4 The equation is difficult to correspond to the previous X-ray diffraction analysis.

p.6 183 Not Fig 1, but Fig. 6.

Response: Thanks to the reviewers for their suggestions on revision, and the unclear points have been revised one by one.

The purpose of studying Ca2+dissolution is to prove that calcined SAAC has hydrolytic activity and can be rehydrated, which has been modified in this paper.

The interlayer water of tobermorite is not lost at a certain temperature point, but gradually lost at 50-300 ℃, which has been explained in the article.

The diffraction peak of tobermorite in SAAC has been supplemented in this paper. The complete diffraction peak of tobermorite can be considered as having high crystallinity.

The interlayer water is lost below 300 ℃. The Si OH bond breaks when calcined at 750 ℃, and the hydroxyl is lost at this time, which has been modified in the article.

The equation has been modified and the text has been supplemented to correspond to the text as much as possible.

5) The conclusion is not refined enough.

Response: The conclusion has been revised and the research conclusions of the paper are listed one by one.

Yours sincerely,

Xiuli Yang

Reviewer 2 Report

The paper is still deficient or needs to be optimized in the following areas.

(1) Please highlight the background of the study, the main work, the novelty and the results of the study in the abstract.

(2) The content of the introduction is too simple. The second paragraph is close to the first one, and it is suggested that it be trimmed and adjusted to one paragraph. Has there been any research conducted by scholars on AAC. If yes, please add the corresponding references and highlight the innovation points; if not, please enrich the research background.

(3) Avoid using pronouns that are highly subjective in your essay (e.g., we, etc.).

(4) The difference between two types of autoclaved aerated concrete? Why choose only SAAC for calcination? Compressive strength test but then chose untreated FAAC?

(5) Lack of part of the experimental process, must provide the relevant part of the work.

(6) How to explain  ‘’It is well known that ..."

(7) What is the basis for choosing "150°C, 500°C, 600°C, 750°C, 825°C, 900°C and 1000°C"?

(8) Lines 112-121: This article is related to methodology, please prove it.

(9) Line 183: Please check the figure number.

(10) Line 212: Please check the reference serial number.

(11) Line 237: Please check the error.

(12) The main conclusions are not prominent and lack transitional content and critical discussion. Attempt to map and correlate between each result to ensure continuity of discussion.

(13) The references are too few, the introduction and methodology both lack specific references, and the arguments are highly subjective. Add at least twenty references.

(14) There are many irregularities in the text. Please check the serial number of each chart and correct it.

Author Response

Response letter

We sincerely thank the editors and reviewers for their suggestions and comments, which we use to improve the quality of the manuscript. The reviewer's comments are displayed in normal font. Our reply is given in italics, and changes to the manuscript are given in blue text.

(1) Please highlight the background of the study, the main work, the novelty and the results of the study in the abstract.

Response: The abstract has been revised to give a general overview of the paper and emphasize that the research focus of this paper is the activation of SAAC.

(2)The content of the introduction is too simple. The second paragraph is close to the first one, and it is suggested that it be trimmed and adjusted to one paragraph. Has there been any research conducted by scholars on AAC. If yes, please add the corresponding references and highlight the innovation points; if not, please enrich the research background.

Response: The introduction has been modified and expanded. The added content is mainly the research status in this field.

(3) Avoid using pronouns that are highly subjective in your essay (e.g., we, etc.).

Response: It has been checked and modified.

(4) The difference between two types of autoclaved aerated concrete? Why choose only SAAC for calcination? Compressive strength test but then chose untreated FAAC?

Response: SAAC and FAAC are two different material systems with different compositions and activation effects. FAAC research will be discussed in another article. In the strength test of activity index in this paper, FAAC was not studied. In the contrast experiment, SAAC calcined at different temperatures was used to replace 30% cement. In addition, a comparative test was carried out with fly ash.

(5) Lack of part of the experimental process, must provide the relevant part of the work.

Response: The description of relevant experimental processes has been supplemented, including the titration of ef-CaO by sucrose method, the measurement of hydration time by conductivity, and the preparation of activity index test block. Some experiments, such as DTA-TG and XRD, are routine standard experiments of materials, and no explanation is needed.

(6) How to explain  ‘’It is well known that ..."

Response: The contents to be explained are supplemented and improved.

(7) What is the basis for choosing "150°C, 500°C, 600°C, 750°C, 825°C, 900°C and 1000°C"?

Response: The selection basis of temperature is supplemented in the paper. The temperature is selected according to the position of exothermic peak and endothermic peak in DTA curve.

(8) Lines 112-121: This article is related to methodology, please prove it. Lines 112-121:

Response: The effective calcium is measured by chemical titration. The detection method and principle of effective calcium have been supplemented in the revised version.

(9) Line 183: Please check the figure number.

Response: The drawing number has been checked and modified.

(10) Line 212: Please check the reference serial number.

Response: The serial numbers of references have been checked and modified

(11) Line 237: Please check the error.

Response: Errors have been checked and modified.

(12) The main conclusions are not prominent and lack transitional content and critical discussion. Attempt to map and correlate between each result to ensure continuity of discussion.

Response: The conclusion has been revised and the research conclusions of the paper are listed one by one.

(13) The references are too few, the introduction and methodology both lack specific references, and the arguments are highly subjective. Add at least twenty references.

Response: References have been added.

(14) There are many irregularities in the text. Please check the serial number of each chart and correct it.

Response: The serial number has been checked and modified.

Yours sincerely,

Xiuli Yang

Reviewer 3 Report

In the attached document my notes

Author Response

Response letter

We sincerely thank the editors and reviewers for their suggestions and comments, which we use to improve the quality of the manuscript. The reviewer's comments are displayed in normal font. Our reply is given in italics, and changes to the manuscript are given in blue text.

1)The abstract must be revised as it does not show the significance or innovation of the research.

Response: The abstract has been revised to give a general overview of the paper and emphasize that the research focus of this paper is the activation of SAAC.

2)The introduction needs to be extended, there are numerous researches related to the topic. For example, "Influence of Waste Tire Particles on Freeze-Thaw Resistance and Impermeability Performance of Waste Tires/Sand-Based Autoclaved Aerated Concrete Composites" provides further results that complement this work.

Response: The introduction has been modified and expanded. The added content is mainly the research status in this field.

3)On the line 35 the word AAC is repeated

Response: Errors have been checked and modified.

4)Materials.

The type of cement and sand used to make the specimens must be added. The test methodology is not included. Equipment used and standards should be added in this section.

Response: The manufacturer of cement and sand used for making samples has supplemented the materials.

5Experimental results and discussion

Figure 3. There are peaks that are not identified. For example: diffractograms e) and d) peak at 30°.

Response: Figure 3 has been modified and the marks of unidentified diffraction peaks have been added.

6)The third equation of the reactions occurring during calcination needs to be revised. is it the rehydration after calcination at 745°C? Between line 104 and 105.

Response: The third equation in the first draft is not a strict equation, which has been deleted in the revised draft. The purpose of this equation is to express that the Tobermorite transition phase can hydrolyze Ca2+ions, which is explained in the revised version.

7)On the line 130: “The SAAC powder calcined at different temperatures was suspended at the concentration of 1g/L.” What size of powder? How SAAC was milled?

Response: SAAC powder is milled with a ball mill and its particle size has been tested. The particle size data of SAAC powder has been supplemented in materials.

8)The type and brand of equipment must be specified. For example, on Line 131: “Conductivity of the aqueous solution was measured continuously for 10 hours with conductivity meter shows the conductivity curve of SAAC suspension calcined at different temperatures.”

Response: The brand and type of experimental equipment have been supplemented in the paper, and the experimental method has been described.

9)On the line 183: “Figure 1. the heat of hydration curve of SAAC calcined at 750℃” it should be figure 6.

Response: Errors have been modified and corrected.

10)On the line 195: “The testing result selects the average compressive strength value of six samples.” The study includes 10 different samples. In the sentence I suppose that it refers to 6 half-prisms of each sample tested in compressive stress. If that is the case, this should be clarified.

Response: There are 10 groups of comparative experiments, and the results of each group are based on the average of 6 samples. It has been explained and supplemented in this article.

11General

There are different letter sizes in the document (For example: line 89 and 90).

On the line 237 the title references have moved.

Response: Errors have been checked and modified.

Yours sincerely,

Xiuli Yang

Round 2

Reviewer 1 Report

The paper has been revised according to the relevant review comments, and the quality of the paper has been improved, but there are still many problems. First of all, the language needs to be further improved, and it is recommended to be revised by professionals. Secondly, in terms of the content of the paper, it is suggested to strengthen the following aspects further:

1.      Strengthen the description of the innovation of the paper in the introduction

2.      It is better to comprehensively introduce materials, experimental procedures, and testing methods in the second part.

3.      3.2. The author believes that the amorphous substance was formed at 750 ℃, but the analysis only from the XRD patterns does not explain the formation of the amorphous phase, which may need to be proved by other means.

4.      In 3.3, can equations 1 and 2 be supported by other evidence? Is there a standard or other source for the measurement method of ef CaO? At the same time, the explanation of calcium ion dissolution is not clear enough. It is also questionable whether describing the ion dissolution process with "hydrophysis" is appropriate.

5.      In 3.4, the author believes that " When the SAAC calcined at 100℃, 500℃, 600℃, 900℃ and 1000℃, its hydration reaction would not occur .", which is too arbitrary, and the key is the change of hydration degree. From the subsequent strength test, it can be seen that the strength change is obvious under different temperature treatments.

6.      In 3.5, it is better to show the overall heat generation in a picture and divide the area. Although the author has analyzed how heat is generated in this section, it is not clear on the whole.

7.      In 3.6, the strength has a great relationship with the particle size of SAAC after heat treatment, but the author has not given relevant data in the paper. The strength results seem useful for heat treatment, but there is a lack of systematic analysis of the mechanism in the paper.

Author Response

请参阅附件。

Reviewer 2 Report

The issues raised have been appropriately revised to make the article more rigorous.

Author Response

responseletter

We sincerely thank the reviewers for their valuable feedback.Thank you for your affirmation.In the revised version, some content has been added and the way of expression has been improved to further improve the quality of the manuscript.The full text has been revised by professionals and the language has been further improved. The content added for the second time and changes are marked in green text.

This article has the following modifications.

1.A paragraph has been added in the introduction, outlining the views and describing the innovation points.

2.In the second part, the experimental method for measuring ef-CaO was added, and the activity index method was adjusted to the second part.

3.A complete hydration heat release diagram has been added.

4.In this paper, the particle size distribution of SAAC calcined at various temperatures is added, and the change of particle diameter of SAAC calcined is analyzed.

5.The accuracy of the expression is modified.

Reviewer 3 Report

Las modificaciones propuestas se han aplicado adecuadamente. El artículo es más coherente.

Author Response

(The authors gave the same response as above.)
